# Electrochemical Properties of PEDOT:PSS/Graphene Conductive Layers in Artificial Sweat

**DOI:** 10.3390/s24010039

**Published:** 2023-12-20

**Authors:** Boriana Tzaneva, Mariya Aleksandrova, Valentin Mateev, Bozhidar Stefanov, Ivo Iliev

**Affiliations:** 1Department of Chemistry, Faculty of Electrical Engineering and Technology, Technical University of Sofia, Kliment Ohridski Blvd., 8, 1000 Sofia, Bulgaria; b.stefanov@tu-sofia.bg; 2Department of Microelectronics, Faculty of Electronic Engineering and Technology, Technical University of Sofia, Kliment Ohridski Blvd., 8, 1000 Sofia, Bulgaria; m_aleksandrova@tu-sofia.bg; 3Department of Electrical Apparatus, Faculty of Electronic Engineering, Technical University of Sofia, Kliment Ohridski Blvd., 8, 1000 Sofia, Bulgaria; vmateev@tu-sofia.bg; 4Department of Electronics, Faculty of Electronic Engineering and Technology, Technical University of Sofia, Kliment Ohridski Blvd., 8, 1000 Sofia, Bulgaria

**Keywords:** conductive polymer, spray coating, artificial sweat, cyclic voltammetry, electrochemical impedance spectroscopy

## Abstract

Electrodes based on PEDOT:PSS are gaining increasing importance as conductive electrodes and functional layers in various sensors and biosensors due to their easy processing and biocompatibility. This study investigates PEDOT:PSS/graphene layers deposited via spray coating on flexible PET substrates. The layers are characterized in terms of their morphology, roughness (via AFM and SEM), and electrochemical properties in artificial sweat using electrochemical impedance spectroscopy (EIS) and cyclic voltammetry (CV). The layers exhibit dominant capacitive behavior at low frequencies, with cut-off frequencies determined for thicker layers at 1 kHz. The equivalent circuit used to fit the EIS data reveals a resistance of about three orders of magnitude higher inside the layer compared to the charge transfer resistance at the solid/liquid interface. The capacitance values determined from the CV curves range from 54.3 to 122.0 mF m^−2^. After 500 CV cycles in a potential window of 1 V (from −0.3 to 0.7 V), capacitance retention for most layers is around 94%, with minimal surface changes being observed in the layers. The results suggest practical applications for PEDOT:PSS/graphene layers, both for high-frequency impedance measurements related to the functioning of individual organs and systems, such as impedance electrocardiography, impedance plethysmography, and respiratory monitoring, and as capacitive electrodes in the low-frequency range, realized as layered PEDOT:PSS/graphene conductive structures for biosignal recording.

## 1. Introduction

The use of organic conductive coatings is of interest for applications in various wearable and stretchable electronic devices, primarily owing to their favorable mechanical properties that ensure the long-term durability of the coating and, consequently, the high reliability of the device during multiple bending instances [1]. This characteristic stands as a significant advantage of conjugated polymers, including polythiophenes, polypyrrole (PPy), and polyaniline (PANI), which address the limitations of rigid metal electrodes that often exhibit brittleness [2]. Among these polymers, Poly(3,4-ethylenedioxythiophene) (PEDOT) stands out due to its chemical stability and tunable conductivity over a broad range. Its versatility has led to applications in photovoltaics, displays, and other optoelectronic devices, benefiting from its optical transparency in addition to electrical conductivity [3].

Despite these advantages, PEDOT faces challenges such as poor solubility and higher resistance compared to conventional electrodes. To address these issues, an additive, poly(4-styrenesulfonate) (PSS), has been introduced. This addition transforms the composite PEDOT:PSS into a soluble form in organic solvents, specifically, in an aqueous dispersion.

PEDOT:PSS-based electrodes have been proposed for use in different sensors and biosensors due to their simple processing and biocompatibility [4]. PEDOT:PSS provides a versatile matrix for incorporating various biological components, such as enzymes and peptides [5]. Several studies have explored the use of PEDOT:PSS in electrochemical sensors for monitoring biological targets like glucose, proteins, and other biomarkers [6,7]. Developing flexible chemical sensors for different analytes often involves modifying or functionalizing the working electrodes with suitable substances to achieve the desired sensitivity, selectivity, and stability. PEDOT:PSS and its composites are known for their chemical stability, which makes them highly desirable among conductive polymers. To tailor the electrical conductivity and work function of the material, various methods can be employed. These include doping with nanoparticles, combining with ultrathin metal layers in a multilayer structure, or forming composites with carbon-based nanomaterials. Introducing polar solvents like glycerol, ethylene glycol, or polyalcohols can induce morphological changes in PEDOT:PSS layers, leading to improved charge transport properties [8]. For example, doping with Ca and Mg has demonstrated a positive impact on the conductivity of PEDOT:PSS by enhancing electron mobility [9]. Moreover, discussions have centered around enhancing conductivity through improvements in crystallinity and the orientation of crystalline domains achieved through specific processing conditions [10].

One particularly effective strategy is the incorporation of metal particles, grids, or nanocoatings between two films of PEDOT:PSS. Examples of reported solutions include using silver (Ag) nanoparticles to form grids with different meshes in combination with PEDOT:PSS [11]. Another approach involves the insertion of gold nanoparticles [12]. Furthermore, using combinations of graphene and multiwall carbon nanotubes has shown not only a significant decrease in resistivity but also improved thermal properties and strain resistivity. These strategies offer ways to enhance the electrical properties and overall performance of PEDOT:PSS-based materials [13,14].

Enhancing electrode performance can be challenging when working with flexible, chemically unstable, and thermally sensitive substrates like polyethylene terephthalate (PET). To address this, a dispersion containing PEDOT:PSS and carbon material are preferred for the direct coating and stress-free post-deposition processing of films. Recently, PEDOT:PSS has been combined with graphene. Graphene plays an essential role in these conductive layers due to its exceptional electrical and mechanical properties. Its incorporation into PEDOT:PSS coatings improves the electrical conductivity and stability of the resulting films [15]. Moreover, the combination of graphene with PEDOT:PSS allows for enhanced dispersion and uniformity, which can contribute to the performance and reliability of devices integrating these materials. The use of graphene in conductive layers offers opportunities for advancements in wearable sensor devices and other practical applications that require reliable and highly conductive non-metallic materials. However, the synthesis of materials and technology has faced limitations when it came to producing an integrated single dispersion with the necessary electrical parameters and stability, leading to complicated deposition techniques and often insufficient thin layer coatings. Examples include graphene foams with a PEDOT:PSS network [16] and a 3D tubular graphene sponge with PEDOT:PSS [17].

New inks based on composites between PEDOT:PSS and graphene dots have advanced, offering opportunities for simpler and more cost-effective deposition technology. Various approaches have been explored for the growth of PEDOT:PSS/graphene films, including electro-polymerization (where PEDOT is electropolymerized on carbon paper-coated graphene [18]), spin-coating [19], and inkjet printing [20]. However, challenges remain concerning the uniformity and agglomeration of these coatings. To enhance uniformity, one strategy involves spray coating on a heated substrate, which prevents graphene aggregation and segregation during subsequent thermal treatments. Another noteworthy variation involves the substrate vibration-assisted spray coating of graphene-doped PEDOT:PSS nanocomposites [21]. This approach improves precursor mixing, drying, and evaporation rates. However, vertical vibration can destabilize the nanocoating. The application of spray coating for PEDOT:PSS/graphene has been demonstrated in the creation of a semitransparent capacitive touch device. This large-scale structure does not necessitate the precise tuning of surface roughness [22].

The electrochemical behavior of PEDOT:PSS/graphene coating has been investigated through potentiostatic electrochemical impedance spectroscopy (EIS) [23,24,25,26,27,28,29], cyclic voltammetry (CV) [26,27,28,29,30], and linear sweep voltammetry (LSV) [27,29]. Despite the similarities in the methods and testing conditions for conductive polymer layers, data regarding the electrochemical stability of PEDOT:PSS coatings are not uniform. Some authors report a reduction in its charging ability with continuous cycling [26], while others assert that PEDOT:PSS film remains stable over a hundred cycles [28,31]. This variability may depend not only on the methods for obtaining PEDOT-based layers but also on the electrolytes used in the testing. It is essential to consider that PEDOT and its derivatives are hydrophilic and involve ions such as Na^+^, Cl^−^, and K^+^ [32,33]. Therefore, their electrochemical behavior depends on the exchange of these ions with the surrounding electrolyte. The most commonly used electrolyte in electrochemical tests is neutral phosphate-buffered saline (PBS) [23,26,27,28,29,34]. PBS contains Na^+^, K^+^, Cl^−^, and H^+^ ions comparable to extracellular fluid, making this solution suitable for short-term in vitro electrode characterization. Single tests have been conducted in other electrolytes such as sodium chloride (0.001–0.1 M NaCl) [24,25], KCl [35], H_2_O_2_, and K_4_Fe(CN)_6_ solutions [30]. However, neither the composition nor the acidity of these solutions closely resembles that of human sweat [36]. For the practical applications of PEDOT:PSS/graphene coatings as functional layers or as the main electrode material in wearable sensor devices, it is crucial to determine their electrochemical behavior in an environment as close as possible to the anticipated operational conditions. Unfortunately, studies on these polymer layers in electrolytes resembling human sweat are lacking in the literature. Unlike the test solutions mentioned above, artificial sweat is a more concentrated electrolyte, with a greater variety of ions and a slightly acidic pH of 4.7, providing a more accurate representation of the electrochemical behavior in an operational environment for wearable sensors.

In this study, we explore the potential utilization of a conductive composite layer consisting of PEDOT:PSS/graphene as an independent electrode material intended for future applications in biosensors. The deposition of PEDOT:PSS/graphene was achieved through spray coating on a polyethylene terephthalate (PET) substrate. The resulting conductive layers underwent morphological and electrical characterization, with crucial electrochemical parameters determined through conventional techniques such as electrochemical impedance spectroscopy (EIS) and cyclic voltammetry (CV). The impact of the deposition conditions, resulting thickness, and roughness on their impedance characteristics and electrochemical stability in artificial sweat has been established.

Overall, the study contributes to the understanding of the principles underlying the utilization of PEDOT:PSS/graphene as an electrode material. The novelty of the work is in the utilization of spray coating for deposition, the characterization of the conductive layers, and the evaluation of their performance and stability in a simulated sweat environment. These findings provide insights into the potential application of the composite layer in biosensors and contribute to the development of wearable and reliable sensor devices. This work is significant as it addresses key challenges in the fabrication, characterization, and performance evaluation of PEDOT:PSS/graphene conductive composite layers for biosensor applications. The findings can guide further research in the field of wearable sensor technology and contribute to the development of innovative and practical sensing devices with enhanced functionalities.

## 2. Materials and Methods

PET flexible substrates, measuring 25 mm by 25 mm, and with a thickness of 270 µm, underwent cleaning with isopropyl alcohol in a supersonic bath, followed by UV–Ozone surface treatment (λ = 265 nm, P = 350 W). The hybrid material, consisting of 1 mg/mL PEDOT:PSS/graphene in dimethylformamide (DMF) solvent (Sigma Aldrich—Merck KGaA, Darmstadt, Germany) (0.2 mg/mL PEDOT:PSS and 1 mg/mL electrochemically exfoliated graphene; 500 Ω/sq, 20 nm film: 80% transmittance), was prepared. Samples were fabricated using a spray coating setup HS-AS18CK Haosheng (Ningbo, China) equipped with a regulating nozzle, model HS-30, having a diameter of 0.01–0.1 mm and a hot plate. The substrate temperature ranged from 90 to 105 °C, controlling the liquid flow rate and film uniformity. The aerosol pressure varied between 2 and 3.6 mbar. Table 1 details the specific deposition conditions, maintaining a constant nozzle-to-substrate distance of 12 cm. Under fixed deposition conditions, film thickness was determined solely by the number of passes. The pre-heated substrate was exposed to the aerosol flow in 5-s pauses, allowing solvent evaporation and preventing material leakage. The tested layers were designated as a sequence of the substrate temperature, the spray pressure, and the number of passes. For example, the indication S90-3.5-10 should be read as a sample obtained at a temperature of 90 °C with a pressure of 3.5 mbar and 10 passes.

Sheet resistance and its deviation across the coated area were measured using a four-point probe setup (FPP 5000), implementing the van der Pauw method. Atomic force microscopy (AFM) analysis was conducted to estimate roughness (Nanosurf, FlexAFM, Liestal, Switzerland). AFM observations were made in at least 3 points, scanning an area of 10 µm^2^ at 512 × 512 resolution, with a 0.2 N m^−1^ soft sample cantilever. Scanning electron microscopy (SEM) was performed using Tescan LYRA (Brno, Czech Republic) to explore the morphology of the coatings before and after electrochemical tests. Quantitative image analysis was carried out in ImageJ 1.8.0. The data on the average size of the PEDOT agglomerate were obtained via the integrated “Analyze particles” plugin on dimension-calibrated SEM images. Image processing was also employed to calculate the PEDOT agglomerate-covered areas for each sample via Sage’s “Color Segmentation” plugin for ImageJ in all cases. The average PEDOT agglomerate diameter was calculated from ImageJ-obtained area data based on the equivalent circle formula.

Electrochemical impedance spectroscopy (EIS) and cyclic voltammetry (CV) were conducted on artificial sweat using a potentiostat–galvanostat PalmSens4 equipped with a frequency response analyzer and using PSTrace 5.9 software. A minimum of 2 series of layers were made under the different spray conditions, and each sample was tested at least at 3 points. Electrochemical tests were carried out in a three-electrode cell with the working electrode composed of the nanocomposite PEDOT:PSS/graphene on a PET substrate and a working area of 0.1 cm^2^. A Pt-plate (1 cm^2^) and an Ag/AgCl electrode in 3.0 M KCl with potential versus the standard hydrogen electrode of 0.21 V were used as the counter electrode and reference electrode, respectively. All potentials in the study are presented with respect to the Ag/AgCl electrode. The electrolyte was prepared based on the work by Naser et al. with a composition of 20 g L^−1^ NaCl, 17.5 g L^−1^ NH_4_Cl, 5 g L^−1^ acetic acid, and 15 g L^−1^
*d*,*l*-lactic acid with the pH adjusted to 4.7 using NaOH [37]. All reagents were analytically pure: NaCl (≥99.0%, CAS №: 7647-14-5), NH_4_Cl (≥99.5%, CAS №: 12125-02-9), acetic acid (≥99.8%, CAS №: 64-19-7), *d*,*l*-lactic acid (CAS №: 50-21-5) and were purchased from AlfaAesar, Karlsruhe, Germany.

EIS tests were conducted at an open circuit potential with an amplitude of 10 mV in a frequency range from 10^5^ to 10^−2^ Hz. Cyclic voltammetry (CV) was employed to determine the voltammetric capacitance (*C*_CV_) of the tested layers by assessing the slope of the linear fit of the dependence of the capacitive current density on the scan rate, which was varied from 0.01 to 1.0 V. The voltammetric charge *Q*_CV_ of the samples was calculated from the area of CV hysteresis using the following formula:(1)$$Q_{CV}=\frac{1}{2v}\int_{E_{1}}^{E_{2}}j(E)dE,$$
where *ν* is the scan rate (0.1 V s^−1^), *E*_1_ and *E*_2_ are the potential window, and *j* is the current density at each potential.

To investigate electrochemical stability, 500 cycles were conducted within an experimentally defined potential window (from −0.3 to 0.7 V) with a scan rate of 0.1 V s^−1^.

## 3. Results and Discussion

### 3.1. Characterization of PEDOT:PSS/Graphene Layers

Key parameters such as thickness, sheet resistance, and roughness were determined for the tested layers, and the arithmetic mean results from the measurements are presented in Table 1. Generally, with an increase in temperature (S90-3.5-10 and S105-3.5-10), the layer thickness and sheet resistance decrease. The increase in spray pressure does not positively influence the thickness; for instance, layers at 3.5 bar (S100-3.5-5) are approximately 20% thinner, with their sheet resistance being twice as high, reaching a value of 111.6 Ω/sq. When comparing data for samples S100-3.5-5 and S105-3.5-10, it is observed that the thickness and resistance of the layers are not linearly correlated with the number of passes. Thus, with twice the number of passes, the resistance decreases slightly for S105-3.5-10, and the thickness does not reach a doubled value. The influence of deposition conditions on the physical and electrical properties of the layers has been investigated and presented in our previous work [38].

AFM topographical images are depicted in Figure 1. The obtained images reveal a relatively smooth surface. The determined values for average roughness obtained through AFM are presented in Table 1. At a large scale (100 µm^2^), a true area increase in the electrode surface is not significant; moreover, it is in the range of 0.2% for the S100-2.0-5 sample, reaching nearly 5% for S90-3.5-10. The specific surface is not increased significantly for that kind of material and deposition technology. This is the picture at a μm scale: at small local areas, the peaks of the surface roughness could increase the specific surface by 30% or more, where, on average, they converge to create the presented values.

Lower spraying temperatures can affect the wetting behavior of the coating solution on the substrate. If the solution does not spread uniformly and wet the surface effectively, it can lead to the formation of irregularities and higher roughness [39]. At lower temperatures, the particles are more likely to stick to the surface in a more localized manner, which leads to a rougher coating. This is because lower temperatures provide less energy for the particles to flow and spread out. In addition, the combination of a lower spraying temperature, higher pressure, and multiple passes can promote the aggregation and phase separation of the PEDOT:PSS and graphene materials in the coating solution. This can lead to the formation of clusters or the uneven distribution of the conductive materials, contributing to higher roughness [21]. It appears that temperature has a more significant impact on the roughness of the coated samples compared to the spraying pressure and number of passes. Achieving similar roughness values can indicate that the coating process is consistent and reproducible within that temperature range, providing a level of control over the resulting surface characteristics. Thus, 100–105 °C can be accepted as an optimal temperature range. In addition, when the substrate is hotter, the particles are more likely to flow and spread out on the surface, which leads to a smoother coating. This is because higher temperatures provide more energy for the particles to overcome the surface tension and flow more easily. Thus, the substrate temperature is the dominant factor affecting the roughness of the coatings in this case.

The variation of the surface roughness is related to the film thickness and the change of the sheet resistance, which can explain the highest value of ΔR_sd_ (57.6 Ω/sq), corresponding to the larger roughness of 35.7. Regarding the dissimilar values of the sheet resistance variation for the similar values of the roughness, a possible reason can be the degree of the homogeneity of the coating across the entire surface area, the presence of agglomerates, as can be seen in Figure 2, and the connectivity and alignment of conductive particles within the coating. The agglomerate distribution, even smoother in their topography, can result in localized areas of higher or lower conductivity, leading to variations in sheet resistance from point to point across the entire area of the samples.

The surface morphology was further examined through SEM (Figure 2).

Regardless of the deposition conditions, PEDOT:PSS/graphene layers exhibit a similar structure. Bright, spherical formations are noticeable on all surfaces, representing agglomerates of PEDOT molecules connected with PSS [36]. PEDOT clusters are distributed with varying uniformity in a light-gray matrix of PSS [36]. A detailed analysis of images from various test samples reveals that the size of PEDOT agglomerates increases by approximately 65% with an increase in pressure from 2.0 to 3.5 bar (comparing S100-2.0-5 and S100-3.5-5) and an increase in substrate temperature from 90 to 105 °C (comparing S90-3.5-10 and S105-3.5-10). Thus, the average radius of PEDOT agglomerates for samples S90-3.5-10 and S100-2.0-5 is around 48 nm; meanwhile, for samples S100-3.5-5 and S100-3.5-5, it is 79.6 nm and 80.8 nm, respectively. These dimensions are approximately twice as large as those reported for the PEDOT:PSS layers obtained through drop-casting [27,40]. In sample S100-3.5-5, PEDOT agglomerates occupy the largest surface area (about 40%), merging into larger structures. However, the agglomeration of PEDOT clusters does not improve the conductivity of the polymer. Sample S100-2.0-5, characterized by the smallest area of PEDOT clusters (about 18%) and, correspondingly, the largest area of free PSS, exhibits the lowest sheet resistance value (Table 1).

### 3.2. Charge Transfer through the PEDOT:PSS/Graphene Layers in Artificial Sweat

The electrochemical properties of PEDOT:PSS/graphene layers were investigated through electrochemical impedance spectroscopy (EIS) and cyclic voltammetry (CV) in a simulated human sweat environment. The primary criteria for evaluating the quality of the tested polymer layers are formulated from the perspective of their use as standalone electrodes, with the requirement not to alter the signal, which is achieved by maintaining a maximally low and frequency-independent impedance over a wide frequency range [41]. In this regard, the Bode plot provides valuable information about how well different frequency input signals are transmitted across the solid–liquid interface [41]. The data obtained from the electrochemical impedance spectroscopy results are graphically represented in Figure 3 as Nyquist and Bode plots.

The complex plane plots of S90-3.5-10, S105-3.5-10, and S100-2.0-5 (Figure 3a) exhibit a significant slope without the presence of a charge transfer semicircle. This indicates pure capacitive behavior and fast ion diffusion. The Nyquist plot for the S100-2.5-5 layer has the form of an open semicircle, demonstrating charge exchange in the low-frequency region. From the Bode plots, the cut-off frequency (*f*_cut-off_) is determined, below which the electrode material transitions from predominantly resistive to capacitive behavior. The value is determined at the intersection point between the resistive and capacitive branches of the impedance and coincides with the phase angle of −45° (Figure 3b) [40]. As the thickness of the polymer layer decreases in the direction from S90-3.5-10 > S105-3.5-10 > S100-2-5 > S100-3.5-5, the critical frequency *f*_cut-off_ progressively shifts to higher values from 0.96 to 3.21 kHz (Table 2), consistent with information in the literature [42]. However, these values can be considered relatively close, suggesting that the layers S90-3.5-10, S105-3.5-10, and S100-2-5 exhibit similar behavior with a frequency-independent signal above 2.5 kHz. Since 1 kHz is a representative frequency in biosignaling, it can be inferred that the layers with a greater thickness—S90-3.5-10 and S105-3.5-10—are more suitable as electrode materials in this regard [28]. Nevertheless, considering that some methods, such as impedance electrocardiography, impedance plethysmography, respiratory monitoring, etc., utilize frequencies in the range of 20–150 kHz, it can be concluded that all tested PEDOT:PSS/graphene-based layers could be deemed suitable, regardless of the deposition conditions.

PEDOT:PSS/graphene layers in artificial sweat exhibit relatively stable capacitive behavior in the frequency range of 10^−2^–10^2^ Hz, where the phase angle remains more negative than −75°. These results align with those described by Mousavi et al. for electrodeposited PEDOT:PSS films [28]. The thinnest layer, S100-3.5-5, besides having a higher *f*_cut-off_, also demonstrates a narrower capacitive region, with the phase angle dropping below −60° at low frequencies. This indicates a parallel occurrence of surface reactions contributing to charge transfer [27].

Various equivalent circuits are employed in the analysis of PEDOT:PSS-based layers, ranging from the simplest series RC circuit [25,26], Randles circuit, *R*_s_(*C*_dl_(*R*_ct_*W*)) [28,43,44], to considerably more complex ones involving multiple RC blocks and Warburg impedance (W) [28,29], depending on the substrates and additives used in the conductive polymer. The best fit for our experimental EIS results was achieved with the equivalent circuit presented in Figure 3a. This circuit is denoted by the description code [*R*_s_([*R*_ct_(*R*_i_*Q*_i_)]*C*_dl_)], where *R_s_* represents the solution resistance. The capacitance *C_dl_* provides the charge at the electrolyte–polymer interface (i.e., the contribution of the double-layer capacitance) in parallel to the impedance block *R*_ct_(*R*_i_*Q*_i_), representing all additional charge transfer mechanisms (different physical processes contributing to charge transfer). The element *R*_ct_ is the charge transfer resistance of the external solid/liquid interface [29]. The internal block (*R*_i_*Q*_i_) reflects various transfer mechanisms within the polymer composite, such as ion flow in the ionic channels in PSS and electrical conductivity in graphene and PEDOT. In the internal parallel block, the capacitance is replaced with a constant phase element (*Q*_i_) since, in the average frequency range, the phase displacement does not reach −90°. Thus, *Q*_i_ reflects the surface nonhomogeneity and porosity of the composite layer, and *R*_i_ is the internal resistance. This equivalent circuit is used in the literature to describe the electrochemical behavior of organic layers containing bipolar molecules that interact to form a multilayered assembly capable of blocking electron charge transfer [45]. Fits based on this equivalent circuit are represented as black lines in Figure 3, and the determined values of the elements are summarized in Table 2. The low chi-squared (χ^2^) test values (below 0.001) indicate good fit quality.

The higher value of *R*_s_ recorded for S90-3.5-10 can be attributed to an elongation of the ionic diffusion paths in the thickened film [43]. The values of *R*_i_ are approximately three orders of magnitude higher than those of *R*_ct_, indicating that the charge transfer through the inside of the composite layer is significantly more hampered than the passage through the external electrolyte–polymer interface.

The higher values of the capacitive components of the impedance for the S90-3.5-10 layer indicate the retention of charges at both the solid/liquid interface and the internal boundaries between different polymer phases and graphene. It is known that capacitance increases with an increase in the active electrochemical surface of the electrode (the total three-dimensional contact surface with higher roughness) or the volumetric capacitance, which is proportional to the thickness of the capacitive layer [25,42]. Therefore, the results obtained for the S90-3.5-10 layer are logical given its greater thickness and roughness, as determined by the deposition conditions.

Cyclic voltammetry was employed to investigate the processes of DC charge transfer through the electrode surface and to determine the potential range where mainly capacitive charging takes place. Figure 4a presents the CV dependencies of the tested layers in artificial sweat during bidirectional potential scanning in the range from −0.3 to 0.7 V vs. Ag/AgCl. This range is experimentally determined as the capacitive potential region, dominated by capacitive behavior characterized by nearly constant positive or negative current in the anodic and cathodic direction of potential scanning, respectively. In this relatively wide potential window (up to 1 V), no faradaic reactions occur, and no current related to a typical charge transfer process flows. At more positive potentials above 0.7 V, the current shows a tendency to increase, especially for the S90-3.5-10 layer, indicating the initiation of the forced oxidative processes of the polymeric components in the composite or of organic acids in an electrolyte. The increase in cathodic current at potentials more negative than −0.2 V is most likely related to a hydrogen evolution reduction reaction catalyzed by graphene in the nanocomposite coating [46].

It is known that the pristine PEDOT:PSS involves both groups PSS^–^Na^+^ and PSS^–^H^+^. Out of these, PSS^–^H^+^ possesses better capability in generating carriers than PSS^–^Na^+^ [47]. At lower pH values of the artificial sweat, the amount of H^+^ ions incorporated into the PSS phase is expected to be higher in thicker layers. This could be one possible explanation for the initiation of cathodic reaction at less negative potentials for the layers obtained after 10 passes.

From the closed hysteresis loop of the CV dependencies, an estimation of the voltammetric charge, *Q*_CV_, which represents the total charge transferred during cyclic scanning, has been made [34]. The calculated values are presented in Table 3. The voltammetric charge can also be used for the experimental assessment of the electrochemically active surface area of the electrode [4,48].

Since the recorded current in the capacitive potential region has a capacitive nature, it is directly proportional to the scan rate *ν* [48]. This dependence was used to experimentally determine the double-layer capacitance (*C*_CV_) from CV dependencies taken at potential scan rates ranging from 0.01 to 1.0 V s^−1^. The capacitive current *j*_c_ is measured in the middle of the capacitive potential interval at *E* = 0.2 V vs. Ag/AgCl. This value is close to the open-circuit potential of all tested layers in artificial sweat, where the state of the equilibrium of the double-layer charging process is attained. The dependencies of *j_c_* vs. the potential scan rate for the PEDOT:PSS/graphene layers are presented in Figure 4b. The steepest curve and, consequently, the highest capacitance *C*_CV_ (Table 3) are observed for S90-3.5-10, while the smallest is for S100-2.0-5, which is in line with the results from EIS and *Q*_CV_ at 0.1 V s^−1^. This method for determining *C*_CV_ is sensitive to the porosity and roughness of the electrode films, which is reflected in the deviation from the linear dependence in the low *ν*-domain [48]. In this regard, it is noteworthy that the linear dependence for the thinnest layers, S100-2.0-5 and S100-2.0-5 (with five passes), is maintained even at a low scan rate of 10 mV s^−1^. However, for the thicker layers, especially for the S90-3.5-10 layer, the linear trend is disrupted, and a new linear segment with a greater slope is formed at scan rates below 100 mV s^−1^. This is an indication of the increased roughness and/or porosity of the layer, forming zones (pores, cracks, etc.) with a much slower ion charging process. In this case, more time is required for the charging of the inner interfaces of the film (such as the ionic channels between polymer macromolecules), which can only be registered at a lower scan rate. With an increase in the scan rate, less accessible zones (pores, cracks, etc.) are progressively excluded from the electrochemical response as *ν* increases [48].

The high capacitance values of PEDOT:PSS/graphene layers demonstrate their potential utility as capacitive electrodes connected to amplification stages implemented through standard instrumental amplifiers rather than transimpedance amplifiers with an ultra-high input resistance (TΩ).

### 3.3. Electrochemical Stability

The electrochemical stability of PEDOT:PSS/graphene conductive films is assessed by potential sweeping at a rate of 0.1 V s^−1^ in the range from −0.3 to 0.7 V in an artificial sweat solution. Figure 5 presents CV cycles for each conductive layer. The most significant change in CV dependencies is observed for S90-3.5-10 (Figure 5a) during the first ten scans in the anodic region at potentials above the open-circuit equilibrium value. The reduction in the area of the anodic peak suggests a decrease in anodic activity after partial oxidation in the initial cycles. Slight changes are observed for S105-3.5-10 and S100-2-5 over 500 cycles.

Nevertheless, the CV area and associated electrochemical properties remain stable between 10 and 500 cycles (Figure 6a), indicating the electrochemical stability of all electrode layers during redox processes in the non-faradaic current window. Similar behavior has been reported for PEDOT:PSS layers deposited using other methods [28,49].

The change in capacitance and capacitance retention over 500 CV cycles is presented in Figure 6. As mentioned earlier, the S90-3.5-10 sample exhibits the most significant decrease in *Q*_CV_ after the first cycle. This could be attributed to more pronounced structural changes occurring during the initial electrochemical loading of the layer. These changes are likely associated with the penetration and saturation of the polymer layer with ions from the electrolyte [28]. It can be expected that this effect becomes more pronounced with an increase in the thickness of the layers, where the contraction/expansion of the polymer caused by the release/incorporation of ions from the electrolyte occurring under cyclic electrical loading is more significant [32]. As a result of the changes in the composite layer, the capacitance of the thickest layer, S90-3.5-10, decreases by 16% after the first 50 cycles. However, no further reduction is observed in the subsequent cycles; on the contrary, the capacitance shows a tendency toward weak recovery. The other three samples with smaller thicknesses demonstrate excellent electrochemical stability, with a reduction in *Q*_CV_ below 6% (Figure 6b).

Following cycling, the surfaces of the tested layers exhibit minor surface changes and reorganization. Figure 7 presents SEM images in the BSE mode of the surface of the S100-2.0-5 layer after 500 cycles in artificial sweat, which are analogous to the observed changes in the other layers. Darker and brighter regions are observed after electrochemical cycling (Figure 7a), opposing the surface before the tests at low magnifications (Figure 2c, inset). At higher magnifications (above 20,000×), the brighter regions show partially exposed PEDOT clusters with a tendency to agglomerate (Figure 7b). These observations may be related to the swelling and partial dissolution of the free PSS phase on a part of the surface of PEDOT:PSS/graphene layers [34]. It is known that electrochemical stimulation (charge-discharge) can also induce volume changes in the electrode material, resulting in mechanical stress at the interfacial boundaries. In this process, significant swelling of PEDOT has been observed during ion exchange with the surrounding electrolytes [32].

## 4. Conclusions

The deposition conditions of nanocomposite layers of PEDOT:PSS/graphene through spray coating influence their electrochemical behavior primarily through the thickness and uniformity of the distribution of PEDOT agglomerates in the layer. The layer with the greatest thickness was obtained at a substrate temperature of 90 °C, 3.5 bar pressure, and 10 passes, demonstrating a cut-off frequency of 1 kHz and stable capacitive behavior at lower frequencies. Thinner layers have a slightly higher cut-off frequency while simultaneously exhibiting better electrochemical stability in artificial sweat under cyclic electrical loading, as observed using cyclic voltammetry.

The characteristics of the investigated layers appear promising for the registration of vital parameters and processes in the human body, particularly in the implementations of sensor structures embedded in wearable systems. Resistive behavior at frequencies above 1 kHz and stable phase characteristics are suitable for measuring impedance changes related to the functioning of individual organs and systems, such as heart activity (impedance electrocardiography), peripheral circulation (impedance plethysmography), and breathing (thoracic impedance measurement). Typical frequencies used for these measurements are in the range of 20–150 kHz.

On the other hand, the stable capacitive behavior of the layered PEDOT:PSS/graphene structures at frequencies of 10^−2^–10^3^ Hz would be a crucial factor in the development of recording devices for biomedical signals (electrocardiogram, electroencephalogram, electrogastrogram, etc.) with a frequency spectrum in the range of 10^−1^–10^2^ Hz. The capacitance values of the structures allow their use as capacitive electrodes connected to amplifier stages implemented through standard instrumental amplifiers instead of transimpedance amplifiers with ultra-high input resistance (TΩ).

A subject of future research is the evaluation of the applicability of capacitive electrodes realized as layered PEDOT:PSS/graphene conductive structures for recording biosignals.

## Figures and Tables

**Figure 1 sensors-24-00039-f001:**
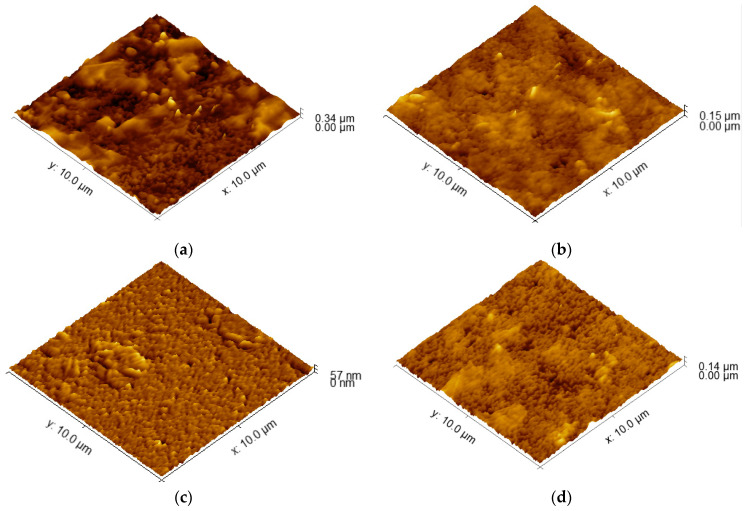
AFM images of the surfaces of PEDOT:PSS/graphene layers for samples: (**a**) S90-3.5-10; (**b**) S105-3.5-10; (**c**) S100-2.0-5; (**d**) S100-3.5-5.

**Figure 2 sensors-24-00039-f002:**
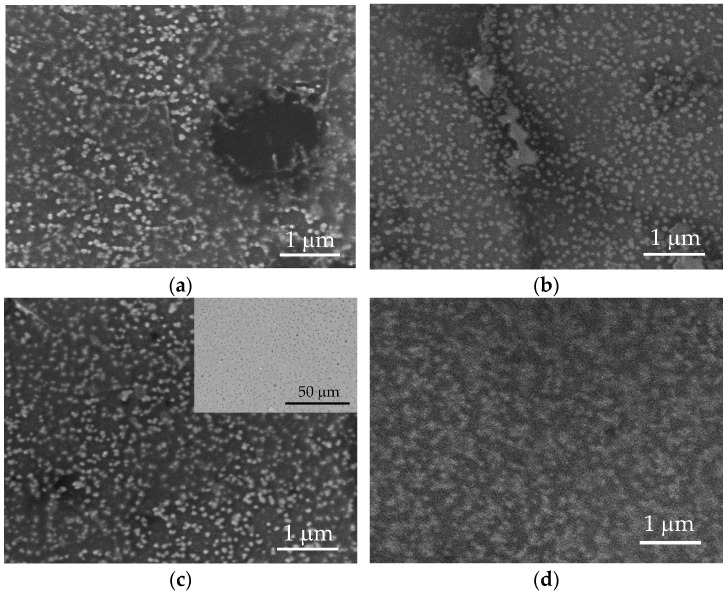
SEM images of surfaces of nanocomposite PEDOT:PSS/graphene deposited under different experimental conditions: (**a**) S90-3.5-10; (**b**) S105-3.5-10; (**c**) S100-2-5 (inset: in BSE mode); and (**d**) S100-3.5-5.

**Figure 3 sensors-24-00039-f003:**
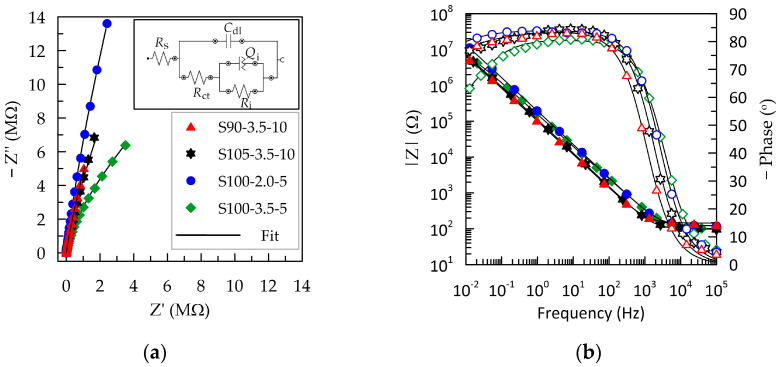
EIS of PEDOT:PSS/graphene layers sprayed under various experimental conditions: (**a**) Nyquist complex plots; (**b**) Bode plots showing magnitude (closed symbols) and phase (open symbols) of the impedance. The legend presented in (**a**) is valid for the entire figure.

**Figure 4 sensors-24-00039-f004:**
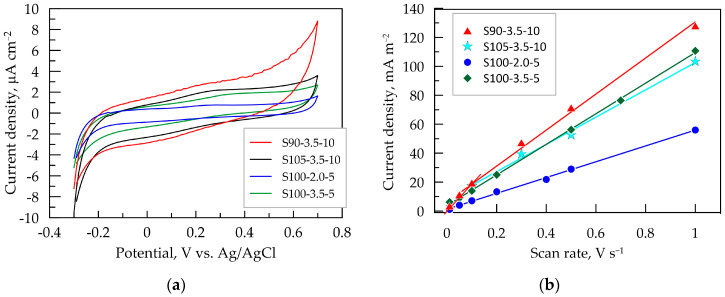
Cyclic voltammetry (CV) results for PEDOT:PSS/graphene layers in artificial sweat: (**a**) CV dependence at 0.1 V s^−^¹; (**b**) dependence of the capacitive voltammetric current on the scan rate of PEDOT:PSS:graphene layers.

**Figure 5 sensors-24-00039-f005:**
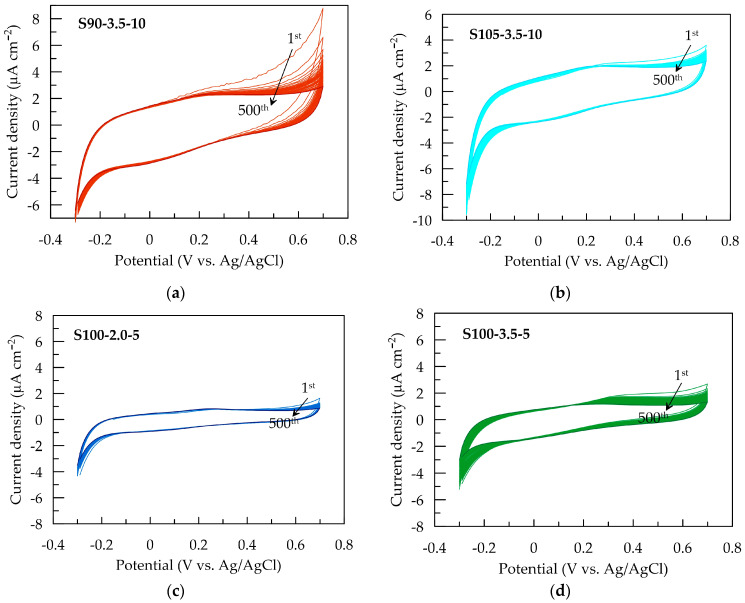
Cyclic voltammograms of PEDOT:PSS/graphene with a scan rate of 100 mV s^−^¹ in artificial sweat for (**a**) S90-3.5-10, (**b**) S105-3.5-10, (**c**) S100-2-5, and (**d**) S100-3.5-5.

**Figure 6 sensors-24-00039-f006:**
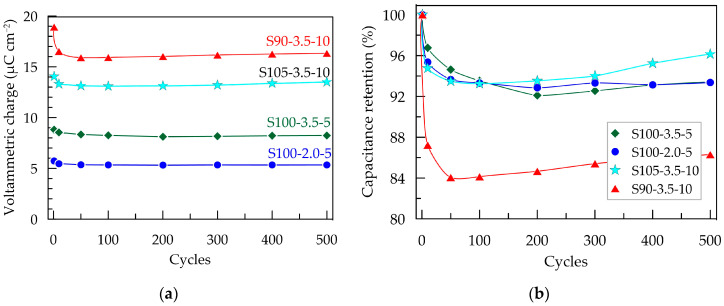
Electrochemical stability of PEDOT:PSS/graphene layers in artificial sweat after 500 CV scans with a scan rate of 100 mV s^−^¹ within a potential window from −0.3 to 0.7 V; (**a**) voltammetric charge; (**b**) capacitance retention.

**Figure 7 sensors-24-00039-f007:**
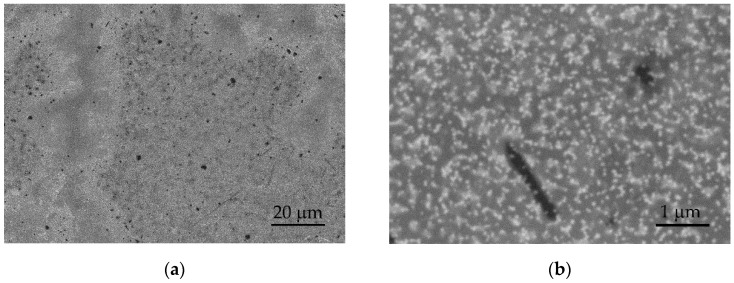
SEM images in BSE mode of the S100-2.0-5 surface after 500 CV cycles at: (**a**) 1000× and (**b**) 20,000× magnification.

**Table 1 sensors-24-00039-t001:** Experimental conditions for spray coating and key properties of PEDOT:PSS/graphene layers.

Sample	Temperature, °C	Pressure, Bar	Number of Passes	Thickness, µm	R_s_, Ω/sq	ΔR_sd_, Ω/sq	Roughness, nm
S90-3.5-10	90	3.5	10	3.25 ± 0.05	117.9	57.6	35.7 ± 6.8
S105-3.5-10	105	3.5	10	1.8 ± 0.05	90.3	29.7	8.3 ± 0.8
S100-2.0-5	100	2.0	5	1.25 ± 0.05	51.3	5.1	9.1 ± 2.3
S100-3.5-5	100	3.5	5	1.00 ± 0.05	111.6	49.4	9.2 ± 0.7

**Table 2 sensors-24-00039-t002:** Data from electrochemical impedance spectra fitting of PEDOT:PSS/graphene layers.

Sample	*f*_cut-off_, kHz	*R*_s_, Ω	*R*_ct,_ Ω	*R*_p_, MΩ	*Q*_i_, µF s^n^	*n*	*C*_dl_, nF	*χ^2^*
S90-3.5-10	0.96	137.8	80.6	93.8	1.23	0.865	641.6	0.0005
S105-3.5-10	1.55	97.8	192	95.7	0.927	0.831	764.4	0.0007
S100-2.0-5	2.31	114.7	30.6	220.7	0.767	0.912	175.2	0.00009
S100-3.5-5	3.21	101	146.9	24.86	0.961	0.82	342.6	0.0003

**Table 3 sensors-24-00039-t003:** Voltammetric charge and differential capacitance values obtained from CV dependencies of PEDOT:PSS/graphene layers.

Sample	*Q*_CV_, µC cm^−2^	*C*_CV_, mF m^−2^
S90-3.5-10	18.93 ± 2.41	122.0 ± 6.2
S105-3.5-10	14.05 ± 1.02	94.2 ± 4.6
S100-2.0-5	5.51 ± 1.14	54.3 ± 2.9
S100-3.5-5	8.82 ± 0.97	105.6 ± 6.3

## Data Availability

Data underlying the results presented in this paper are not publicly available at this time but may be obtained from the authors upon reasonable request.

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
