# Peer review of "Electrochemical Properties of PEDOT:PSS/Graphene Conductive Layers in Artificial Sweat"

_sensors, 2023, doi:10.3390/s24010039_

Round 1

Reviewer 1 Report

Comments and Suggestions for Authors

The manuscript describes the electrochemical properties of PEDOT:PSS/Graphene conductive layers, which is deposited via spray coating on flexible PET substrates. The layers are sufficiently characterized in morphology, roughness, electrochemical properties and electrochemical stability. This work has some practical significance, as the electrochemical characterization is processed in artificial sweat. However, some minor adjustments need to be done before publication in sensors. Several issues are as follows:

1.       The conductive layers are mentioned as PEDOT:PSS/Graphene in the introduction part, but in the latter part of the manuscript the layers are mentioned as PEDOT:PSS/Gr.

2.       The introduction part should highlight the principles and innovations of the work, to describe more about the importance of graphene in the conductive layers, to demonstrate more date to illustrate the differences between using artificial sweat and other electrolytes solutions.

3.       Some of the references in this article are not new.

Comments on the Quality of English Language

none

Reviewer 2 Report

Comments and Suggestions for Authors

In the manuscript entitled “Electrochemical Properties of PEDOT:PSS/Graphene Conductive Layers in Artificial Sweat”, the authors have studied in detail the electrochemical properties of a spray painted PEDOT:PSS/Graphene polyethylene terephthalate in artificial sweat. The study results have been well presented. However, a few changes to the manuscript will benefit the scientific community. 

1.     The introduction in the present form is not easy to read. Using shorter sentences with well-connected information will help the readers to a great extent. 

2.     In line 343 “Fig. 4a” has to be replaced with “Fig. 5a”.

3.     Information on the instruments used is not complete. For instance, the make and model of instrument used for spray painting along with the information on the nozzle and the AFM will be helpful.

4.     The resolution of AFM images in figure 1 needs to be improved. 

Comments on the Quality of English Language

The introduction in the present form is not easy to read. Using shorter sentences with well-connected information will help the readers to a great extent. 

Reviewer 3 Report

Comments and Suggestions for Authors

The paper entitled "Electrochemical Properties of PEDOT:PSS/Graphene Conductive Layers in Artificial Sweat" can become suitable for publication after following amendments.

1. A graphical abstract must be included in the current study to enhance the readers' comprehension of the research findings.

2. Reaction mechanism should be discussed.

3. How many replications were performed for the experiments? In addition, statistical evaluations should be carried out for experimental results. Experimental error bars should be illustrated in figures.

4. The measurement of specific surface area of the modified electrodes should be elaborated.

5. There is no discussion about the influence for pH of buffer solution.

6. CAS numbers and purity of sall the used chemicals should be provided.

Comments on the Quality of English Language

Moderate editing of English language required.

Reviewer 4 Report

Comments and Suggestions for Authors

The paper entitled “Electrochemical properties of PEDOT:PSS/graphene conductive layers in artificial sweat” by Tzaneva et al. described the investigation of electrochemical properties of graphene and PEDOT:PSS based electrode in artificial sweat.  This result is interesting and can be accepted but several things need to be considered before being published as follows:

1.       The paragraph in lines 81-83 is too short. The authors can combine it with the previous paragraph or it can still be elaborated further.

2.       How to prepare an artificial sweat?

3.       The author needs to explain and discuss the ΔRsd value in the results and discussion section.

4.       Why sample S90-3.5-10 shows the highest roughness value among other samples when it was investigated with AFM?

5.       Why there is a small difference in roughness values observed in the 3 samples (S105-3.5-10, S100-2.0-5, and S100-3.5-5)?

6.       Did the authors also analyze different areas/spots when investigated with the AFM technique in order to evaluate the consistency of roughness value from each sample?

7.       Why there are no results about EDX mapping for 4 samples to show the distribution of elements on the surface?

8.       Why does only Figure 2 show the inset on a 50 μM scale? What about the other samples?

9.       From SEM analysis in Figure 2, how to distinguish between PEDOT clusters with graphene layers?

10.    How to obtain the average radius of PEDOT agglomerates in SEM figures? Is there any histogram data about the calculation of the radius of PEDOT agglomerates?

11.    Why the Nyquist plot in Figure 3 does not show a semicircle?

12.    The authors need to put a legend in Figure 3b.

13.    Line 248, 10-2 – 102 Hz need to be superscript

14.    Why did the authors not try to characterize or study the electrochemical behavior of the electrode using a redox probe molecule?

15.    Why does the sample of S100-2-0.5 show the lowest capacitive currents compared to other sample electrodes as shown in Figure 5?

16.    Is there any possibility that one compound consisting of an artificial sweat will undergo an electrochemical redox reaction when cyclic voltammetry experiment was performed?

Comments on the Quality of English Language

The manuscript is well-written
